# The Growth Path of Agricultural Labor Productivity in China: A Latent Growth Curve Model at the Prefectural Level [†]

**Peng Bin [1,2,*] and Marco Vassallo [3]**

[1]  School of International Studies, University of Trento, Via Tommaso Gar 14, Trento 38122, Italy
[2]  College of Public Administration, Huazhong Agricultural University, Wuhan 430070, China
[3]  Agricultural Research Council (CRA), Via Ardeatina 546, Rome 00178, Italy; vassallo@inran.it
[*]  Correspondence: peng.bin@unitn.it; Tel.: +39-046-128-3132
[†]  An initial version of this paper is presented in the 133rd EAAE seminar "Developing Integrated and Reliable Modelling Tools for Agricultural and Environmental Policy Analysis", 15–16 June 2013, Chania, Crete, Greece.

**Abstract:** Given the shrinking proportion of agriculture output and the growing mobility of the labor force in China, how agricultural labor productivity develops has become an increasingly attractive topic for researchers and policy makers. This study aims to depict the development trajectory of agricultural labor productivity in China after its WTO entry. Based on a balanced panel data containing 287 Chinese prefectures from 2000 to 2013, this study applies the Latent Growth Curve Model (LGCM) and finds that the agricultural labor productivity follows a piecewise growth path with two breaking points in the years of 2004 and 2009. This may stem from some exogenous stimulus, such as supporting policies launched in the breaking years. Further statistical analysis shows an expanding gap of agricultural labor productivity among different Chinese prefectures.

**Keywords:** agricultural labor productivity; growth trend; divergence; agricultural reforms

**JEL Classification:** O13; O47; Q18

## 1. Introduction

As a new emerging economy with a large population to feed, how to improve agricultural productivity has always been a key point for food security and social stability in China. It had been a top priority to provide people with adequate food and clothing in the early stage of China's industrialization and urbanization. This demand has been continually upgraded with the rapid socioeconomic development of the country after its market liberalization. The market reform brings farmers not only more access to production goods, but also additional job opportunities in the other industries. Meanwhile, the development in science and technology releases a great amount of the agricultural labor force and boosts productivity. It is commonly accepted that the Total Factor Productivity (TFP) is contributed as a main incentive to the growth of agricultural output in China. Even though its agricultural input stagnated during the 1990s, agricultural output kept increasing with a respectable annual growth of TFP around 2% (Fan, 1997 [1]). Cao and Birchenall (2013) [2,3] further conclude that the agricultural labor input has been decreasing at an annual rate of 5%, and agricultural TFP has been growing by 6.5% in the past few decades. Wang et al., (2013) [4] observe a tendency of increasing regional disparity given that the coastal regions hold faster growth rates of agricultural TFP. There is no doubt that the increase of agricultural productivity facilitates the labor surplus moving to other sectors, but the growing liquidity of the agricultural labor force constrains

the growth of agricultural productivity conversely (Rozelle, 1999 [5]). The agricultural productivity gap across countries has been investigated in rich detail in the study of Gollin et al., (2002) [6]. Gollin et al., (2014 [7,8]) further examine the cross-country agricultural productivity gap in their recent studies. They conclude the existence of a cross-country agricultural productivity gap. They investigate the extent of the gap in terms of agricultural labor productivity by taking the measures of sector inputs and outputs into consideration (2014 [7]) and confirm the existence of large productivity differences in the agricultural sector by micro and macro data on productivity in various grain products. It is commonly accepted that the agricultural labor productivity increases in China along with its socioeconomic development. However, how this growth path is characterized still remains unclear, given the complex interactions of labor and productivity in agriculture. Statistics show that the nominal output and input of China's agriculture kept increasing during the recent decades. However, it is hard to capture this tendency in terms of the real values, since the index of agricultural input and output fluctuates dramatically (see Appendix A). This paradox blurs the real growth path of agricultural productivity, especially the agricultural labor productivity, considering the high fluidity of the labor resource from agriculture to industry in China during the recent years.

When it comes to the growth pattern of the Chinese economy during the last decade, we cannot ignore the lash of the global economic crisis in any terms. To cope with the crisis, the Chinese government launched an investment project "4-Trillion-Yuan Stimulus Package" ($586 billion) in 2008–2009. The distribution of this stimulus package is as follows: the investments to housing guarantees are 0.4 trillion RMB, to rural construction are 37 million RMB, to energy conservation and emissions reduction are 21 million RMB, to infrastructure development are 1.5 trillion RMB, to social services are 15 million RMB, to industrial restructuring are 37 million RMB and to post-disaster reconstruction of Wenchuan are one trillion RMB (World Bank, 2010 [9]). The Chinese government intended to expand domestic demand and improve people's livelihood through this stimulus project. This stimulus has been published along with doubts and queries from the very beginning. Some economists stated that this government move would sabotage the Chinese economic structure in the long run and mislead the economy. Hence, how agricultural labor productivity grows under this complex macro environment could be an interesting perspective to evaluate this stimulus package.

It is hard to depict the development of agricultural labor productivity in China without paving the agricultural policy background. The series of reforms can definitely be attributed as a significant factor to promote agricultural labor productivity. Taking the Household Responsibility System (HRS) for instance, this reform greatly pumps up the pulse of farmers' productivity and, hence, stimulates agriculture production in the whole of China. This positive effect of HRS has been quantified to initiate an annual growth rate of 5%–10% in farm output and productivity at the initial stage during 1978 and 1985 (Lin, 1992 [10]). However, the Chinese government gave more priorities to urban construction and industry development during the 1980s and 1990s. This policy preference brought large-scale land expropriation, city expansion and labor migration and, hence, jeopardized the agricultural production and further aggravated urban-rural disparity. To balance the inequality and deal with the great challenges brought up by entering the WTO, a series of new agriculture-oriented policies centered on the reforms of agricultural tax and grain subsidy had been initiated to protect agricultural production since 2000. The central government implemented the regulation of direct grain subsidy at the national level in 2004. This year is also the starting year to reduce the agricultural taxes. Many research works focused on the conversion of the governmental role from an agricultural taxer to a subsidizer and the following changes along with this conversion (Huang et al., 2004 [11]; Gale et al., 2005 [12]; Huang et al., 2011 [13]; Jin et al., 2010 [14]). The tax abolition came out as an effective reform in increasing rural income and agricultural production via the boost of fixed-capital input, rather than labor input and the improvement of labor productivity instead of capital efficiency, and the gradual reduction of the agricultural tax rate launched initially promoted agricultural productivity as the positive effects of governmental supports to farmers' incentives in agricultural production. The direct subsidy also increased rural income and had no distortions on producer decisions, so it was not in

contrast to the WTO green-box policy (Huang et al., 2011 [13]). The implications of those reforms were more symbolic, and their effects on rural income and grain production were marginal, contributing 2%–4% of the value of agricultural production [1] (Gale et al., 2005 [12]). However, the direct grain subsidy has been questioned in the last few years. For instance, The Economist once questioned the government intervention in farming in China. The article claims the subsidy as the "wrong direction" and suggests to introduce the market mechanism into agricultural (The Economist, 2015 [15]). The Chinese government started to make adjustments to the scheme of the direct subsidy in 2015, aiming to protect the farmland productivity and keep the grain production at an appropriate scale.

Understanding the dynamic trajectory of agricultural labor productivity has been of long-time interest for researchers and policy makers. In order to understand how agricultural labor productivity develops under the influences of such intricate exogenous factors, this study uses the Latent Growth Curve Model (LGCM) and convergence estimation to depict the growth trajectory of agricultural labor productivity at the prefectural level. It also conducts some further analysis on mapping the regional performances in terms of the agricultural labor productivity. This is the first time the LGCM has been used to estimate China's growth pattern of agricultural labor productivity at the prefectural level. The results contribute to the current knowledge. The paper is structured as follows: Section 2 will describe the data and methodology; the results of the LGCM will be presented and discussed in Sections 3 and 4, respectively; and Section 5 will conclude.

## 2. Methodology and Data

### 2.1. Latent Growth Curve Modeling

The Latent Growth Curve Model (LGCM) is a bold and innovative application of the Structural Equation Model (SEM) to analyze the changes in repeated measures over time both at the aggregate and the individual level (Preacher, 2010 [16]). The main rationale of using the LGCM approach lies in defining, hence capturing, the aspects of change throughout a latent variable analysis and thereby gain all of the benefits of SEM, since LGCM is a special case of SEM. These aspects of change are latent in nature, meaning that they have an unobservable nature, and their existence is evidenced by the interrelations among the observed repeated measures. These interrelations are indeed treated through a multiple-indicator Confirmatory Factor Analysis (CFA) (Brown, 2006) [17], under the usual SEM framework, in order to model the aforementioned aspects of change as factors. These factors reflect the dynamics of change over time in terms of the means, variances and covariance of individual differences that, in turn, may be also explained through the introduction of external, time or non-time invariant, variables (Hancock and Lawrence, 2006 [18]). In this respect, the capacity of SEM to simultaneously estimate and handle shifts in the variance, covariance and mean structure over time permits LGCM to apply more information in the repeated measures variables than traditional methods (i.e., ANOVA, MANOVA, ANCOVA, MANCOVA, auto-regressive, cross-lagged multiple regression) with no premise that all of the individuals change at the same rate and have the same fluctuations at each time. The only requirement of LGCM is that the individual growth paths follow the same functional form, for simplicity's sake, which has to be hypothesized as linear at the beginning, but other functions can be modeled (i.e., quadratic, cubic, exponential, etc.) if this is not the case.

Figure 1 describes the path diagram of a typical unconditional LGCM, where $Y_0$ is the agricultural labor productivity at the initial time (year 2000), $Y_1$ is the same variable measured at the second time point (year 2001), until $Y_{13}$ for the year 2013. The two growth factors are defined as the intercept $\alpha$ and the slope $\beta$. The former represents the amount of the measured variable Y at the initial point that is defined as $Y_0$ in this case, whereas the latter represents how much that individual score changes, for each time, after and in reference to the initial point. It is actually noteworthy that the loadings $\lambda$ of the

---

[1]　The increase in grain production during 2004 was due primarily to a 30-percent increase in grain prices.

intercept are all constrained to 1 as they equally influence all of the repeated measures across all of the waves of assessment (Bollen and Curran, 2006 [19]). On the other hand, the loadings of the slope are constrained to an ordinal sequence, as they reflect an equally-spaced unit of time series between assessments and an initial linearity form of the trajectory. The notation is typical of SEM, where the latent variables are enclosed in circles or ellipses, observed variables in rectangular boxes and the error terms $\varepsilon_i$ free of lines. The single-headed arrow from latent to observed variable is the impact of the former on the latter, whilst the double-headed arrow between the two latent variables represents the covariance between them. The trajectory equations are therefore composed of two levels (Bollen and Curran, 2006 [19]) as follows:

$$Y_{it} = \alpha_i + \lambda_t \beta_i + \varepsilon_{it} \tag{1}$$

$$\alpha_i = Mean\ \alpha_i + Var\ \alpha_i$$

$$\beta_i = Mean\ \beta_i + Var\ \beta_i \tag{2}$$

where $Y_{it}$ represents the agricultural labor productivity of the each prefectural city *(i)* in each year *(t)*, $\alpha_i$ represents the initial level of agro labor productivity at the year 2000, $\beta_i$ represents how much individual changes over each time interval, $\lambda_i$ represents the loadings, $\varepsilon_{it}$ represent measurement errors, as it is reasonable to assume that the unexplained variability by the two growth factors may exist in the repeated measures.

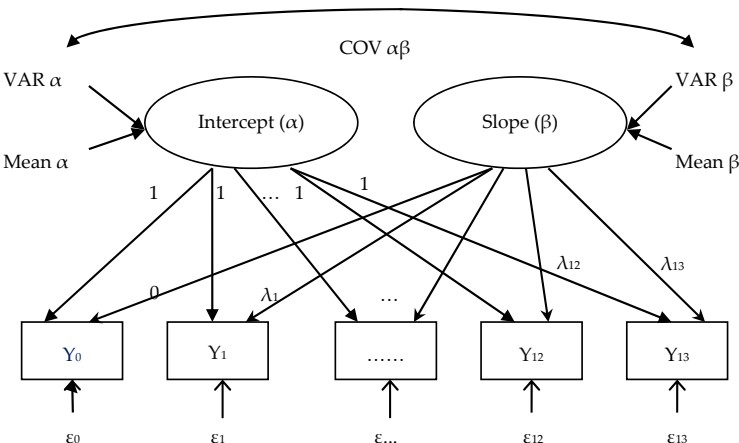

**Figure 1.** Conceptual path diagram.

## 2.2. The Dataset: China's Agricultural Labor Productivity 2000–2013

The LGCM requires a large data sample, in which case, provincial data (only 31 provinces in China) are too small to fit the model, so we choose to use data at the prefectural level [2]. The China City Statistical Yearbooks (2001–2014) are used for our data source. Given the data availability at this level, we use the conception of macro agriculture and calculate the agricultural labor productivity by the quotient of agricultural output and employment [3]. After cleaning the outliers and missing data, there are 279 prefectural cities left [4]. Therefore, the sample size is large enough to apply

---

2   The China City Statistical Yearbook reflects the socioeconomic conditions of the main prefectural cities (or provincial cities) in China. Here, the city is a definition of administrative zoning, a prefectural level, rather than the urban areas. Autonomous prefectures are excluded in this yearbook.

3   We use the output and employment data of macro agriculture, the first sector, as a proxy to agriculture, since they are the only available data at this level to calculate agricultural labor productivity.

4   In order to keep the authenticity of the original data, we delete five samples with missing data and three outliers (Qingyang, Hengyang and Zhongshan), which were only 8 out of 287 initial observations. Tibet was deleted because of the missing values.

LGCM with Maximum Likelihood (ML) estimation with its robust correction for non-normality (i.e., robust maximum likelihood; Satorra and Bentler, 2001 [20]), since the variables $Y_t$ are found to be non-normally distributed (the detailed result is not presented for the sake of brevity, but can be acquired from the first author). We choose 2000 as the starting year mainly due to China's entry into the WTO. The accession to the global market certainly influences the pattern of agricultural production and resource allocation. To cope with the new challenges after joining the WTO, the Chinese government started to shift its focus to agriculture. Considering the importance of WTO to the agriculture sector, we choose the period from 2000 to 2013 as the observation period. The data are deflated with the base year 2000. Since there is no price index at the prefectural level, we use the provincial price indices to deflate the nominal series. Hence, all of the prefectures within each province are deflated by the same index. Although suboptimal, this helps to preserve the comparability in the two estimations.

*2.3. LGCM Processing*

Once fixing the dataset, a preliminary assessment about the changing trends of several random individuals is tested in order to make a theoretical understanding of the trajectory nature and reasonable hypotheses on its functional form (e.g., linear, quadratic, cubic). The free-loading strategy is applied as an exploratory strategy to individualize what type of trajectory our repeated measures are having over the waves of time. The constraint of the first loading as 0 and the last as 1 aims to better interpret the middle loadings as changing proportions and, thus, to explore the form of the trajectory. The unspecified curve LGCM can be expressed with matrix algebra as follows:

$$
\begin{bmatrix} Y_0 \\ Y_1 \\ Y_2 \\ \vdots \\ Y_{13} \end{bmatrix} = \begin{bmatrix} 1 & \lambda_0 \\ 1 & \lambda_1 \\ 1 & \lambda_2 \\ \vdots & \vdots \\ 1 & \lambda_{13} \end{bmatrix} \begin{bmatrix} \alpha \\ \beta \end{bmatrix} + \begin{bmatrix} \varepsilon_0 \\ \varepsilon_1 \\ \varepsilon_2 \\ \vdots \\ \varepsilon_{13} \end{bmatrix} = \begin{bmatrix} 1 & 0 \\ 1 & \lambda_1 \\ 1 & \lambda_2 \\ \vdots & \vdots \\ 1 & 1 \end{bmatrix} \begin{bmatrix} \alpha \\ \beta \end{bmatrix} + \begin{bmatrix} \varepsilon_0 \\ \varepsilon_1 \\ \varepsilon_2 \\ \vdots \\ \varepsilon_{13} \end{bmatrix} \tag{3}
$$

A linear model is first tested by fixing the loadings of the slope with a linear sequence of numbers as follows:

$$
\begin{bmatrix} Y_0 \\ Y_1 \\ Y_2 \\ \vdots \\ Y_{13} \end{bmatrix} = \begin{bmatrix} 1 & \lambda_0 \\ 1 & \lambda_1 \\ 1 & \lambda_2 \\ \vdots & \vdots \\ 1 & \lambda_{13} \end{bmatrix} \begin{bmatrix} \alpha \\ \beta \end{bmatrix} + \begin{bmatrix} \varepsilon_0 \\ \varepsilon_1 \\ \varepsilon_2 \\ \vdots \\ \varepsilon_{13} \end{bmatrix} = \begin{bmatrix} 1 & 0 \\ 1 & 1 \\ 1 & 2 \\ \vdots & \vdots \\ 1 & 13 \end{bmatrix} \begin{bmatrix} \alpha \\ \beta \end{bmatrix} + \begin{bmatrix} \varepsilon_0 \\ \varepsilon_1 \\ \varepsilon_2 \\ \vdots \\ \varepsilon_{13} \end{bmatrix} \tag{4}
$$

Then two latent variables $\beta_2$ and $\beta_3$ are added to hypothesize the quadratic and cubic curves, respectively, as the linear model does not fit well. The quadratic LGCM is:

$$
\begin{bmatrix} Y_0 \\ Y_1 \\ Y_2 \\ \vdots \\ Y_{13} \end{bmatrix} = \begin{bmatrix} 1 & \lambda_0 & \lambda_0^2 \\ 1 & \lambda_1 & \lambda_1^2 \\ 1 & \lambda_2 & \lambda_2^2 \\ \vdots & \vdots & \vdots \\ 1 & \lambda_{13} & \lambda_{13}^2 \end{bmatrix} \begin{bmatrix} \alpha \\ \beta_1 \\ \beta_2 \end{bmatrix} + \begin{bmatrix} \varepsilon_0 \\ \varepsilon_1 \\ \varepsilon_2 \\ \vdots \\ \varepsilon_{13} \end{bmatrix} = \begin{bmatrix} 1 & 0 & 0 \\ 1 & 1 & 1 \\ 1 & 2 & 4 \\ \vdots & \vdots & \vdots \\ 1 & 13 & 169 \end{bmatrix} \begin{bmatrix} \alpha \\ \beta_1 \\ \beta_2 \end{bmatrix} + \begin{bmatrix} \varepsilon_0 \\ \varepsilon_1 \\ \varepsilon_2 \\ \vdots \\ \varepsilon_{13} \end{bmatrix} \tag{5}
$$

The cubic LGCM is:

$$
\begin{bmatrix} Y_0 \\ Y_1 \\ Y_2 \\ \vdots \\ Y_{13} \end{bmatrix} = \begin{bmatrix} 1 & \lambda_0 & \lambda_0^2 \\ 1 & \lambda_1 & \lambda_1^2 \\ 1 & \lambda_2 & \lambda_2^2 \\ \vdots & \vdots & \vdots \\ 1 & \lambda_{13} & \lambda_{13}^2 \end{bmatrix} \begin{bmatrix} \alpha \\ \beta_1 \\ \beta_2 \end{bmatrix} + \begin{bmatrix} \varepsilon_0 \\ \varepsilon_1 \\ \varepsilon_2 \\ \vdots \\ \varepsilon_{13} \end{bmatrix} = \begin{bmatrix} 1 & 0 & 0 \\ 1 & 1 & 1 \\ 1 & 2 & 4 \\ \vdots & \vdots & \vdots \\ 1 & 13 & 169 \end{bmatrix} \begin{bmatrix} \alpha \\ \beta_1 \\ \beta_2 \end{bmatrix} + \begin{bmatrix} \varepsilon_0 \\ \varepsilon_1 \\ \varepsilon_2 \\ \vdots \\ \varepsilon_{13} \end{bmatrix} \tag{6}
$$

Finally, a piecewise linear LGCM (Bollen and Curran, 2006 [19]) is hypothesized as an alternative strategy for dealing with non-linear trends. We determine two breaking points by examining the unspecified curve unstandardized estimates and error covariance (see Appendix B). One is the national agricultural subsidy reform in 2004, and the other is the 2008–2009 Chinese economic stimulus plan (the "4-Trillion-Yuan Stimulus Package", see the article in *China Daily*). Hence, this model can be postulated based on a piecewise strategy setting 2004 and 2008 as breaking points [5], which divide the whole trend into three parts represented by three linear slopes $\beta_1$, $\beta_2$ and $\beta_3$. The function can be expressed with freezing the loadings in the level of growth at the time points of 2004 and 2008 that respectively constitutes a new point of departure for the second growth factor $\beta_2$ and the third factor $\beta_3$. The function is as follows:

$$
\begin{bmatrix} Y_0 \\ Y_1 \\ Y_2 \\ \vdots \\ Y_{13} \end{bmatrix} = \begin{bmatrix} 1 & \lambda_0 & \lambda_0 & \lambda_0 \\ 1 & \lambda_1 & \lambda_1 & \lambda_1 \\ 1 & \lambda_2 & \lambda_2 & \lambda_2 \\ 1 & \lambda_3 & \lambda_3 & \lambda_3 \\ 1 & \lambda_4 & \lambda_4 & \lambda_4 \\ 1 & \lambda_5 & \lambda_5 & \lambda_5 \\ \vdots & \vdots & \vdots & \vdots \\ 1 & \lambda_{08} & \lambda_{08} & \lambda_{08} \\ 1 & \lambda_{09} & \lambda_{09} & \lambda_{09} \\ \vdots & \vdots & \vdots & \vdots \\ 1 & \lambda_{13} & \lambda_{13} & \lambda_{13} \end{bmatrix} \begin{bmatrix} \alpha \\ \beta_1 \\ \beta_2 \\ \beta_3 \end{bmatrix} + \begin{bmatrix} \varepsilon_0 \\ \varepsilon_1 \\ \varepsilon_2 \\ \vdots \\ \varepsilon_{13} \end{bmatrix} = \begin{bmatrix} 1 & 0 & 0 & 0 \\ 1 & 1 & 0 & 0 \\ 1 & 2 & 0 & 0 \\ 1 & 3 & 0 & 0 \\ 1 & 4 & 0 & 0 \\ 1 & 4 & 1 & 0 \\ \vdots & \vdots & \vdots & \vdots \\ 1 & 4 & 4 & 0 \\ 1 & 4 & 4 & 1 \\ \vdots & \vdots & \vdots & \vdots \\ 1 & 4 & 4 & 5 \end{bmatrix} \begin{bmatrix} \alpha \\ \beta_1 \\ \beta_2 \\ \beta_3 \end{bmatrix} + \begin{bmatrix} \varepsilon_0 \\ \varepsilon_1 \\ \varepsilon_2 \\ \vdots \\ \varepsilon_{13} \end{bmatrix} \quad (7)
$$

## 3. Results: China's Trajectory of Agricultural Labor Productivity

### 3.1. The Estimation of Unspecific LGCMs

This study starts with an explorative strategy hypothesizing an unspecified curve trajectory as a reflection of free-estimated factor loadings (Bollen and Curran, 2006 [19]) on the dataset. For the sake of brevity, we only present the unspecified curve unstandardized estimates with free estimated factor loadings with constraining $\lambda_{00} = 0$ and $\lambda_{13} = 1$ and freeing errors between AP03 and AP04 and between AP09 and AP10 [6] (see Figure 2, where AP stands for the agricultural labor productivity Y).

The free loadings signify the cumulative proportions of change of agricultural labor productivity occurring from the first year (2000) to the last year (2010). We can get the cumulative proportions of total change across the time period. It is indicated that 3% of the total change in agricultural labor productivity occurred between 2000 and 2001; 5% of the total change in agricultural labor productivity occurred between 2000 and 2002; 14% of the total change in agricultural labor productivity occurred between 2000 and 2003, and so forth. Figure 3 shows the trend of changing rates during 2000–2013. It is actually noteworthy that the trend of changing rates increases too rapidly to be a linear curve. The changing rates are fluctuant, in other words, the agricultural labor productivity increases faster in some years, while it gets slower in some other years. Several dramatic increases happened during the period, putting forward our initial idea to further explore about what type of exogenous factors might spur the sharp increases to result in a discontinued piecewise trend.

Table 1 verifies our estimation that a significant difference exists in the initial levels of agricultural labor productivity among all of the sample prefectures, or in other words, all of the prefectural cities started with different initial levels of agricultural labor productivity in 2000, as expected. The mean vectors of independent variables (i.e., growth factors) show that the average of the initial level of agricultural labor productivity in 2000 is 762,000 Yuan, and the average annual growth is 3,771,000 Yuan, indicating a substantial increase of agricultural labor productivity. The variances show a

---

[5]   The project of "4-Trillion-Yuan Stimulus Package" (US$586 billion) is firstly proposed by the Chinese government in November 2008, and the formal implementation is in 2009, when the State Council has issued ten measures to expand domestic demand.

[6]   The decreasing in chi-square is higher than the one between AP09 and AP11 (see Appendix B).

significant individual difference of 7,097,000 Yuan existing among prefectural cities around the mean value (762,000 Yuan) in the initial year 2000, or briefly speaking, all of the prefectural cities started at different initial levels of agricultural labor productivity in 2000. Similarly, subsequent significant and even higher individual differences existed in the growth of agricultural labor productivity in the next few years, indicating that all of the prefectures got more dispersed during their growing processes. Furthermore, the positive and significant correlation (0.40) further signifies that a higher starting level of agricultural labor productivity in 2000 is associated with a larger increment across the whole period. In other words, a prefectural city with a higher initial level of agricultural labor productivity developed at a faster rate, while one with a lower initial level grew at a relatively slower rate.

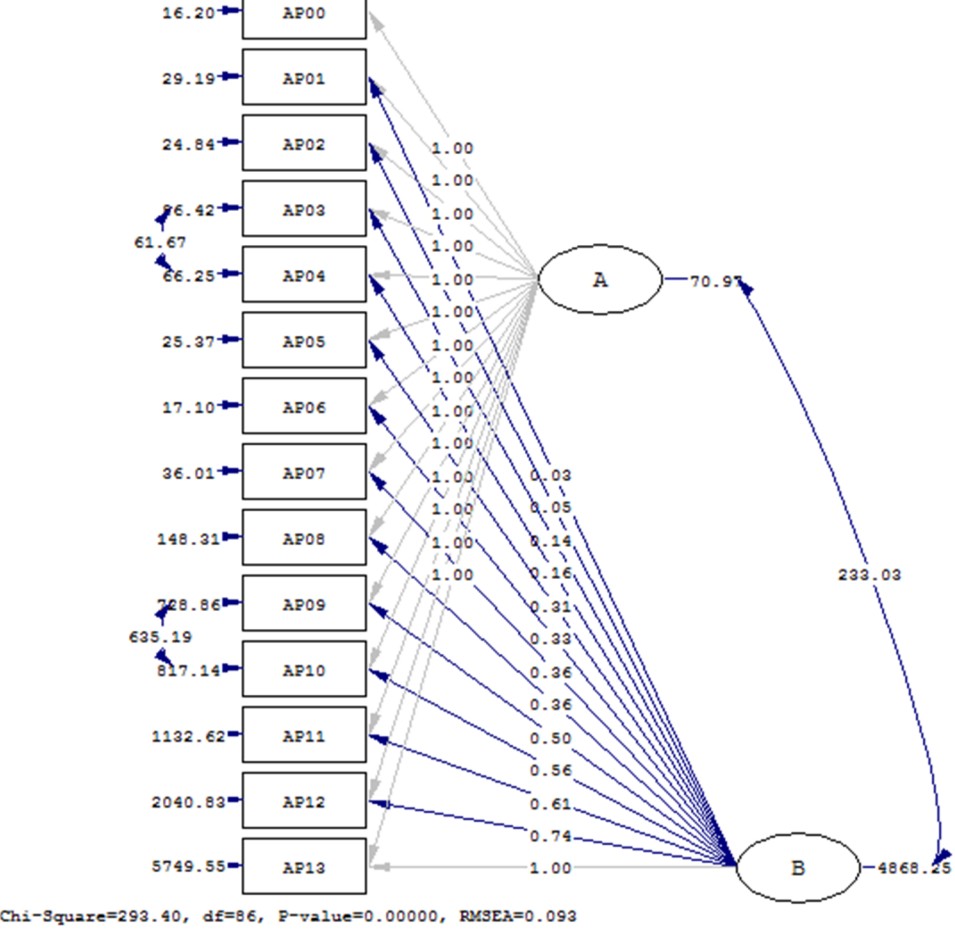

**Figure 2.** The estimation of unspecific Latent Growth Curve Model (LGCM) with free loadings ($\lambda_0 = 0$, $\lambda_{13} = 1$). AP, agricultural labor productivity; RMSEA, Root Mean Square Error of Approximation. Source: LISREL (SSI-Scientific Software International, Inc.) output with application of the data from the Chinese City Statistical Yearbooks (2001–2014) [21].

The proportion of factor loadings of the free-loading strategy reveals that the trend is nonlinear. As a matter of fact, we would find unsatisfactory diagnostics [7] if we hypothesized a linear trend, especially with Root Mean Square Error of Approximation (RMSEA) and Standardized Root Mean Squared Residual (SRMR) as follows: normal theory weighted least squares chi-square = 3185.82

---

[7]　RMSEA with values equal to or less than 0.05 were considered a good fit (Hu and Bentler, 1999 [22]), in the range between 0.05 and 0.08 marginal and greater than 0.10 a poor fit (Browne and Cudeck, 1993 [23]); SRMR should be below 0.09 in good models (Hu and Bentler, 1999 [22]).

($p$ = 0.0); Satorra–Bentler scaled chi-square = 405.65 ($p$ = 0.0); RMSEA = 0.10; 90 percent confidence interval for RMSEA = (0.094; 0.12); standardized RMR = 0.15. Therefore, we confirm that the trajectory of agricultural labor productivity in China from 2000 to 2013 is a non-linear trend. Actually, LGCM fits better with the quadratic model, where the RMSEA is 0.075. However, the low variances and covariance found indicate that the estimation of the quadratic model is still not strong enough, even though the shape of the trajectory of agricultural labor productivity in China may be approximated to a quadratic trend. The cubic model, as tested further, has a slightly better goodness-of-fit than the quadratic one, with a lower RMSEA value of 0.073, but non-significant variances and covariance, implying that the trajectory of agricultural labor productivity may follow a cubic curve, but still it seems an imprecise description of the trend (detailed results on quadratic and cubic models are not reported to preserve space, but they can be requested from the first author).

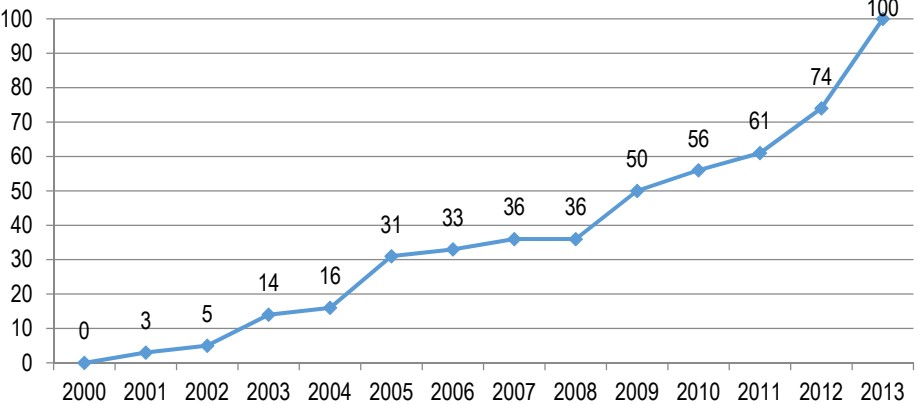

**Figure 3.** The changing rates of agricultural productivity during 2000–2013 (%). Source: data are from the China City Statistical Yearbooks (2001–2014) [21].

**Table 1.** Estimated parameters and *t*-values of free-loading LGCM.

| Parameter | Variances | | Covariance | Correlation | Means | |
|---|---|---|---|---|---|---|
| | **Var ($\alpha$)** | **Var ($\beta$)** | **Cov ($\alpha$, $\beta$)** | **Corr ($\alpha$, $\beta$)** | **$\mu$ ($\alpha$)** | **$\mu$ ($\beta$)** |
| **Estimate** | 70.97 | 4868.25 | 233.03 | 0.40 | 7.62 | 37.71 |
| *t*-values | 4.26 | 3.68 | 3.42 | 3.42 | 13.86 | 5.54 |

Note: *t*-values < |2| are not significant. The unit is one hundred thousand (100,000) Yuan, the Chinese currency RMB (1 Yuan = 0.15 dollars). The observed variables have been rescaled for computational purposes.

### 3.2. The Estimation of the Piecewise Model

Since the exact trend of how agricultural labor productivity changed is not clear enough from the previous models, we hypothesize our nonlinear trajectory with a piecewise linear model, which is still a way to work out nonlinearity trajectories over time. We can clearly observe high error covariance between 2003 and 2004 and between 2008 and 2010 from the free-loading LGCM in Figure 2 and sharp increases of changing rates in 2004 and 2008 in Figure 3. These observations give us an initial idea that something might occur in 2004 and 2008 as some exogenous factors driving the aggregate growth path of agricultural labor productivity far away from a linear trend. To fix the breaking time points more precisely, we checked the modification indices from the free-loading LGCM trajectory depicted in Figure 2 (see Appendix B).

The biggest decrease of chi-square in the years of 2004 and 2008 made it reasonable to hypothesize the years 2004 and 2008 as the points of departure for a second growth factor with freezing the growth attained before (Bollen and Curran, 2006 [19]; Hancock and Lawrence, 2006 [18]) as depicted in Figure 4.

From an empirical point of view, we could also estimate that there might be some exogenous factors having strong influences on the growth trend of agricultural labor productivity, making the year 2004 and the year 2008 considered as two breaking points to make the linear trend discontinuous. In fact, the year 2004 happened to be the year of the direct subsidy for grain launched in the whole nation and the agricultural tax reduction and exemption initiated in China. These two reforms constitute the main body of the new agriculture-oriented policy reform, which have significantly boosted the farmers' enthusiasm in agricultural production and further had effects on the agricultural labor productivity. The year 2008 is the first year of the implementation of the 4-Trillion-Yuan Stimulus Package. This project attempts to buffer against the global financial crisis through heavy investments in various fields, one of which is rural infrastructure and rural livelihoods. As a consequence, it seems both theoretically- and empirically-reasonable to apply the piecewise LGCM strategy with two exogenous factors occurring in 2004 and 2008, as this may also provide further understanding of the new agriculture-oriented policy reforms in China.

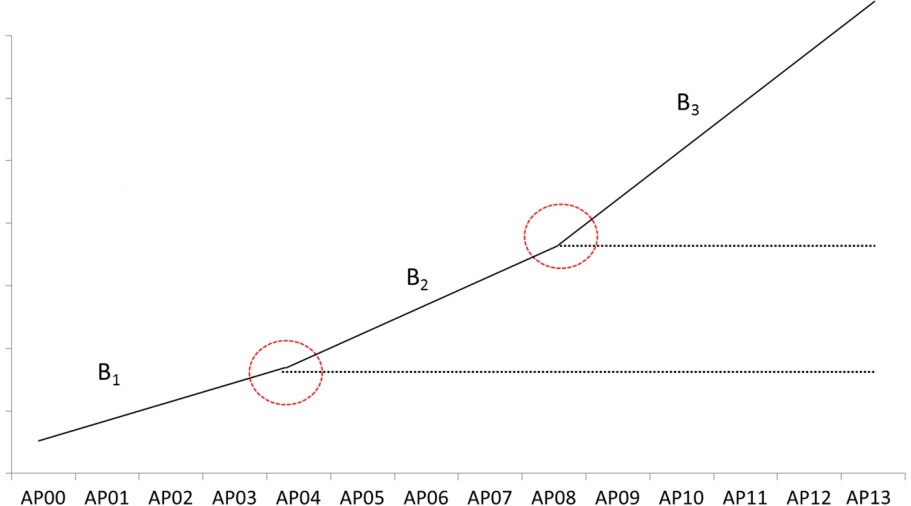

**Figure 4.** The hypothesized growth path. Note: the vertical axis indicates the growth of agricultural labor productivity.

In Figure 4, the solid lines depict the hypothesized new trend, whereas the dotted lines indicate that the initial hypothesized linear mechanism "freezes" the change up to the transition points (2004 and 2008) from which the new trends depart. If the model fit well, the exogenous factors happening in 2004 and 2008 effectively changed the directions of the initial linearity, making the whole trend non-linear. By doing so, the piecewise LGCM is a strategy to deal with the non-linearity of a repeated-measure across waves of time.

The piecewise model fits the data well (normal theory weighted least squares chi-square = 821.28 ($p = 0.0$); Satorra–Bentler scaled chi-square = 137.58 ($p = 0.0$); RMSEA = 0.044; 90 percent confidence interval for RMSEA = (0.029; 0.058); standardized RMR = 0.12), and Table 2 shows that both variances and covariance are significant. It also verifies our previous explorative analysis in the free-loading unspecific model that the initial levels of agricultural labor productivity of all of the sample prefectures are different and correlated with their different growing rates. The piecewise model provides us a further understanding that the prefectural growths of agricultural labor productivity get even more dispersed after the breaking points 2004 and 2008.

**Table 2.** Estimated parameters and *t*-values of the piecewise latent curve model.

| Parameter | Variances | | | | Means | | | |
|---|---|---|---|---|---|---|---|---|
| | $(\alpha)$ | $(\beta_1)$ | $(\beta_2)$ | $(\beta_3)$ | $(\alpha)$ | $(\beta_1)$ | $(\beta_2)$ | $(\beta_3)$ |
| **Estimate** | 71.48 | 13.86 | 19.61 | 69.52 | 7.52 | 1.56 | 2.58 | 6.28 |
| *t*-values | 5.02 | 2.57 | 3.91 | 5.44 | 14.84 | 7.01 | 9.26 | 11.17 |
| | **Covariance** | | | | | | | |
| Parameter | $(\alpha, \beta_1)$ | $(\alpha, \beta_2)$ | $(\alpha, \beta_3)$ | $(\beta_1, \beta_2)$ | | $(\beta_1, \beta_3)$ | | $(\beta_2, \beta_3)$ |
| **Estimate** | 7.27 | 14.57 | 27.69 | 5.58 | | 8.40 | | 27.62 |
| *t*-values | 2.15 | 3.32 | 5.01 | 2.94 | | 1.99 | | 4.29 |
| | **Correlations** | | | | | | | |
| Parameter | $(\alpha, \beta_1)$ | $(\alpha, \beta_2)$ | $(\alpha, \beta_3)$ | $(\beta_1, \beta_2)$ | | $(\beta_1, \beta_3)$ | | $(\beta_2, \beta_3)$ |
| **Estimate** | 0.23 | 0.39 | 0.39 | 0.34 | | 0.27 | | 0.75 |
| *t*-values | 2.15 | 3.32 | 5.01 | 2.94 | | 1.99 | | 4.29 |

Note: *t*-values < |2| are not significant.

We can conclude more specifically that all of the prefectural cities started at an average initial level of 752,000 Yuan [8] of agricultural labor productivity and had an average annual increment of 156,000 Yuan in the first phase from 2000 to 2004. This increment then became 258,000 Yuan for each year in the second phase from 2005 to 2008. A much higher level of annual increment (628,000 Yuan) made the trend more discontinued in the last phase from 2009 to 2013. This indicates that the gaps among different prefectural cities on agricultural labor productivity are further enlarged after 2004 and 2008. Interestingly, it is noteworthy that the covariance related to the growth factors is positive and significant. Therefore, we can deduce that the higher initial levels of agricultural labor productivity are associated with faster growing rates during the whole period. The correlations, however, are more significant in the second and third phase after 2004 and 2008 given their higher correlation values. This implies that the prefectural cities with higher agricultural labor productivities grow faster after 2004 and 2008, and vice versa. Hence, we can infer that the 2004 and 2008 events accelerate the developing disparity.

In order to further understand the inclination or tilt of the prefectural trajectories, we calculated the Relative Gradient (RG) (Hancock and Choi, 2006 [24]) for a non-central standard normal distribution N (RG, 1) (Table 3). We can observe that 66.28% of the estimated slopes $\beta_1$ are positive and 33.72% negative; 71.99% of the estimated slopes $\beta_2$ are positive and 28.01% negative; and 77.3% of the estimated slopes $\beta_3$ are positive and 22.7% negative [9]. The positive slopes indicate increasing trajectories, whereas the negative slopes indicate decreasing trajectories. We can conclude that in the first phase (2000–2004), there are 66.28% of Chinese prefectures with positive growth rates in terms of agricultural labor productivity, whereas 33.72% of prefectures with a negative growth rate; in the second phase (2005–2008), there are 71.99% of Chinese prefectures with positive growths rates and 28.01% with negative growth rates; in the third phase (2009–2013), 77.3% of Chinese prefectures grow positively in terms of agricultural labor productivity, and 22.7% grow negatively. The increasing proportions of

---

[8]    This rescaled value of initial average level of agricultural labor productivity is closer to the actual mean value (832,464 Yuan) in 2000, compared with the previous overestimated initial level 905,000 Yuan in the unspecific model, indicating a better goodness-of-fit of the piecewise model to some extent.

[9]    The Relative Gradient (RG) is a measure of the general inclination or tilt of the trajectories. It is the ratio of mean and standard deviation (Hancock and Choi, 2006 [18]). For a non-central standard normal distribution N (RG, 1), the expected proportion above 0 will be: (a) RG ($\beta_1$) = 0.42, normal distribution probability = 0.6628, thus 66.28% of the estimated slopes $\beta_1$ are positive and 33.72% are negative; (b) RG ($\beta_2$) = 0.58, normal distribution probability = 0.7199, thus 72.24% of the estimated slopes $\beta_2$ are positive and 28.01% are negative; (c) RG ($\beta_3$) = 0.75, normal distribution probability = 0.7730, thus 77.30% of the estimated slopes $\beta_3$ are positive and 22.70% are negative.

prefectures (from 66.28% to 71.99% to 77.3%) show that more prefectures obtain further improvements in terms of their agricultural labor productivities after the two breaking points.

**Table 3.** Relative gradients and normal distribution probabilities.

| Period | Relative Gradient (RG) | Normal Distribution Probability N (RG, 1) (%) | |
|---|---|---|---|
| | | Proportion of Prefectures with Positive Increase | Proportion of Prefectures with Negative Increase |
| $\beta_1$ 2000–2004 | 0.42 | 66.28 | 33.72 |
| $\beta_2$ 2005–2008 | 0.58 | 71.99 | 28.01 |
| $\beta_3$ 2009–2013 | 0.75 | 77.30 | 22.70 |

Source: author's calculation.

### 3.3. The Convergence Estimation of China's Agricultural Labor Productivity

The convergence estimation aims to better illustrate how the agricultural labor productivity evolves the cross-sections. There are two basic types to estimate convergence: one is σ-convergence, which signifies that the cross-section gap of income per capita is reducing in the long run; the other is β-convergence, which reflects a negative correlation between the initial level and the growth rate of income per capita; that is to say, the poorer economies grow faster. The measures of σ-convergence and β-convergence can be well documented in the previous textbook studies (Baumol, 1986 [25]; Barro, 1991 [26]; Barro and Sala-i-Martin, 1991 [27], 1995 [28]; Sala-i-Martin, 1996 [29,30]; Boyle and McCarthy, 1999 [31]; Furceri, 2005 [32]) that are employed in this part [10], and the results are shown in Table 4.

**Table 4.** σ-Convergence and β-convergence estimation of agricultural labor productivity.

| Period | 2000–2013 | 2000–2004 | 2005–2008 | 2009–2013 |
|---|---|---|---|---|
| σ | 13.67 | 2.68 | 1.46 | 2.01 |
| **σ-Convergence** | no σ-convergence | no σ-convergence | no σ-convergence | no σ-convergence |
| β | 0.63 | 1.37 | 0.37 | 0.46 |
| *(t)* | (15.67) | (14.90) | (4.46) | (4.29) |
| **Adj-$R^2$** | 0.3334 | 0.3929 | 0.0296 | 0.0263 |
| **β-Convergence** | non-significant | non-significant | no β-convergence | no β-convergence |

Source: authors' calculation. See Appendix C for the estimation functions of σ- and β-convergence.

Table 4 shows that σ-convergence does not exist at each time interval, which means that the gap of agricultural labor productivity across Chinese regions has enlarged during each time interval.

On the contrary, β-convergence exhibits a different scenario. In the integral period 2000–2013, β-convergence can be observed, but is not very significant, indicating that there is a certain amount of prefectures breaking the rule of β-convergence that lower initial levels of agricultural labor productivity grow faster. The same conclusion can be made for the first period 2000–2004. However, the trend of β-convergence does not exist in the second and third periods (2005–2008 and 2009–2013), which implies that all of the Chinese regions develop at their own paces in terms of agricultural labor productivity in the later stages. Figure 5 displays the trend of β-convergence in each time period. The results of convergence estimation echo the previous analysis based on the piecewise model.

---

10  The elaboration of σ-convergence and β-convergence is omitted here for the sake of brevity (please see Sala-i-Martin (1996) [33] and Barro and Sala-i-Martin (1992) [34] for a reference). The measuring functions are presented in Appendix C.

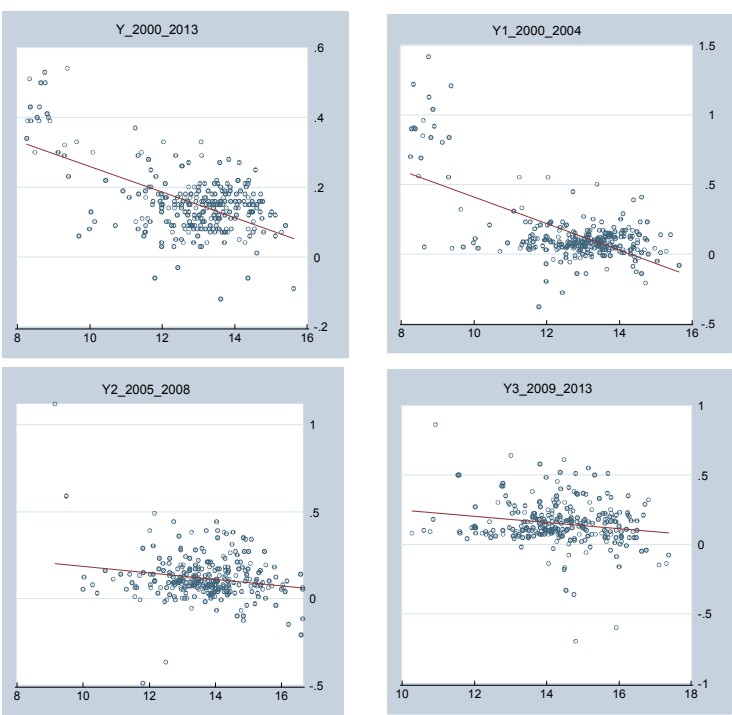

**Figure 5.** Scatter plots of β-convergence estimation. Source: authors' calculation.

## 4. Discussions

### 4.1. The Breaking Year of 2004

In the analysis of the changing trajectory, the year of 2004 was taken as a stepping stone since the government officially launched a series of policies to promote the agricultural tax reform, when the agricultural tax was abolished and the direct subsidy policy was established for farmers in order to increase their incomes and further protect agricultural production. The government conversions from urban-oriented to rural-oriented and from industrial-oriented to agriculture-oriented, greatly improve agricultural production. Many researchers claimed this reform as a significant driver in agricultural production, since it provides obvious welfare to the farmers to encourage their motivations of grain production (Guo and Zhao, 2010 [35]; Li, 2007 [36]; Du, 2011 [37]). Even though quantifying the effects of this reform to agricultural labor productivity is beyond this study, we can still infer that this reform boosts agricultural labor productivity, but brings an expanding gap among different regions as a side effect.

Besides the reform, this breaking point can also be characterized by the labor migration. After relaxing the Hukou system with an experimental one in the new decade, rural labors were allowed more freedom to migrate to non-farming sectors. A series of training programs and beneficial policies for the migrated rural labors were launched during 2002–2004 [11], further encouraging the surplus of agricultural labor to shift to other sectors. Many debates focus on whether China reaches the Lewis

---

[11] In early 2002, the State Council issued the "No. 2 Document of 2002", setting out four principles for labor migration: fair treatment, reasonable guidance, improved management and better services. In 2003, the "No. 1 Document" of the State Council Office drew from these four principles the commitments of abolishing unfair restrictions on rural labors seeking for temporary or permanent employment in urban areas and providing more guaranties in law contracts of payment and healthcare, living conditions, education for their children and training programs. In 2004, a document on the improvement of health services, the prevention of work-related illness and the provision of the treatment of work-related illness among migrant workers was issued, and later in that year, a further document underscoring the necessity for work-injury insurance for migrant laborers was issued to be provided by employers and enterprises, especially in high risk industries, such as construction, mining, etc.

turning point during that period (Cai, 2010 [38]; Flesher et al., 2011 [39]; Zhang et al., 2011 [40]) and find a sharp increase of the wages in the non-agricultural sectors in 2003. A common view is that this wage increase attracts more labors shifting from agriculture to other sectors and, hence, brings labor migration to a peak in 2004. Therefore, it is reasonable that the value of agricultural labor productivity increased sharply in that year to form a piecewise trend, as we calculated by the division of agricultural output [12] and labor force.

### 4.2. The Breaking Point of 2008

This indeed had positive effects to spur domestic demand and support the economic recovery from the global crisis for the country (Fardoust et al., 2012 [41]). However, this fiscal stimulus has been questioned and criticized for its side effects. Under the macro environment, local governments had blindly expanded their investments and production for the sake of their political performances, which planted hidden troubles for the overcapacity and local debt problem (Wong, 2011 [42]). Naughton (2008) [43] has pointed out the most essential problem of the stimulus package, which is the contradiction between the strategy of the stimulus package and the long-term demand in the Chinese economy. To be precise, the key of this stimulus package mainly relies on swelling the government investment, whereas the Chinese economic growth in the long run actually requires improving household income and consumption and further completing the economic transition smoothly. The investment to rural livelihood and infrastructure, no doubt, is beneficial to the welfare of rural households and agricultural production. The construction of rural infrastructure can improve the response abilities to natural disasters and production capacity. However, this study also shows that the disparity of agricultural productivity does not decrease after the stimulus package. In fact, the cross-section gap of agricultural productivity has been enlarged after launching the fiscal stimulus.

### 4.3. Mapping China on Regional Disparities of Agricultural Labor Productivity

To better depict how agricultural labor productivity of different regions developed under the circumstances of this agriculture-oriented policy reform, we mapped the Chinese provinces according to their average annual growth rates of agricultural labor productivity (Figure 6 and Table 5).

It is interesting that the group of fast-growing regions is not concentrated to the east-coast belt as we generally accepted in the analysis of some other indicators of economic regionalization (Bin, et al., 2012 [44]). Instead, many inland regions, such as Hubei, Hunan, Jiangxi, Liaoning and Jilin, which happened to be the experimental pilots of the agriculture-oriented policy reform [13], are included in the fast-growing group, whereas some eastern regions developed at a relatively slow rate on agricultural labor productivity. Considering the potential factors analyzed before, it might be concluded as follows: first, the main grain areas could get more benefits from the agricultural tax and subsidy reforms given the pilot reform initiated in these provinces; second, the inland regions were commonly accepted as exporting agro labor, which had been largely shifting to other sectors and urban areas; third, the local fiscal plans are the vehicles of the stimulus package; hence, the different investment bundles across different regions diverge the growth paths of agricultural labor productivity. This new map provides

---

[12]　The agricultural output has been increasing in aggregate during this new decade, which we have already shown in Figure 1. As far as we are concerned, it may be attributed to the TFP: the continuous improvements of agro science and technology further drive up agricultural production. Interestingly, this viewpoint can also be connected with the first potential factor of agro policy reform. Owing to the Chinese government attaching higher importance to agriculture in the guideline of the new decade, large investments had been put into research and development and, therefore, created a series of breakthroughs in agro science and technology.

[13]　The experimental reform of the agricultural tax reform started in 2002, covering Hebei, Inner Mongolia, Heilongjiang, Jilin, Jiangxi, Shandong, Henan, Hubei, Hunan, Chongqing, Sichuan, Guizhou, Shaanxi, Gansu, Qinghai and Ningxia as pilots, according to their own governmental finances and agriculture conditions; the experimental reform of the direct subsidy for grain also started in 2002, covering Anhui, Jilin, Hunan, Hubei, Henan, Liaoning, Inner Mongolia, Jiangxi and Hebei, the nine main grain production areas, as pilots. As we checked, most of the prefectures in these regions indeed grew at a relatively faster rate after the formal enactment of the agricultural reform.

suggestive information for policy makers and researchers to distinguish the fast-growing group in agricultural labor productivity with the common definition of development in China and, thus, to establish targeted follow-up policies in the future. It is also noteworthy that for some less developed regions, such as Guizhou and Shaanxi, slow-growing productivity may be due to their immature social facility to support the launch of the relative policies and reforms, as far as we are concerned. Therefore, the government has to put more efforts toward improving rural infrastructure construction and agricultural technical equipment in these regions, as well as to coordinate urbanization with rural development and industrialization with agricultural modernization, given that they are still in the economic transition phase.

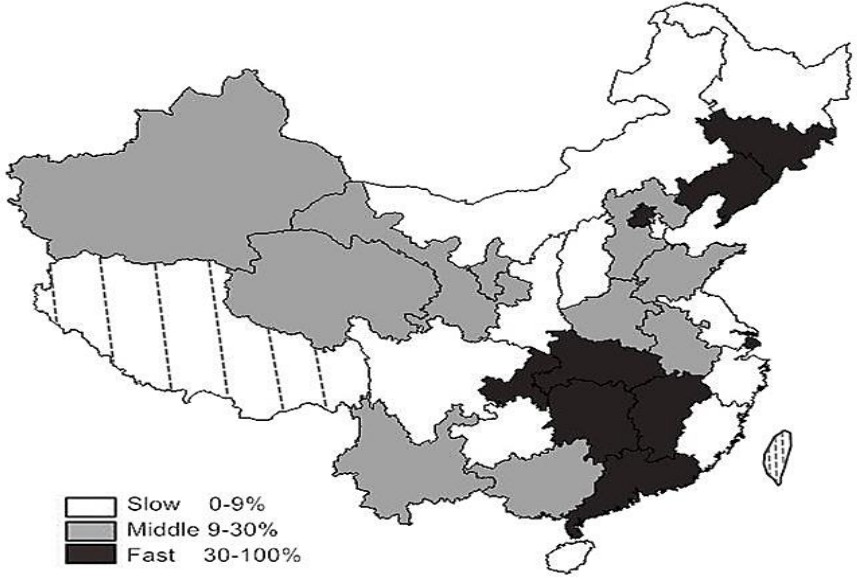

**Figure 6.** Mapping: annual growth of agricultural labor productivity 2000–2013. Source: GIS output based on the data from China City Statistical Yearbooks and China Statistical Yearbooks (2001–2014). Note: see Appendix D for the regional names.

**Table 5.** Annual growth of agricultural labor productivity 2000–2013.

| Groups | Provinces |
|---|---|
| Slow | Heilongjiang, Inner Mongolia, Shanxi, Shaanxi, Sichuan, Guizhou, Tianjin, Jiangsu, Zhejiang, Fujian, Hainan |
| Middle | Hebei, Shandong, Henan, Anhui, Ningxia, Gansu, Xinjiang, Qinghai, Yunnan, Guizhou |
| Fast | Jilin, Liaoning, Beijing, Shanghai, Hubei, Hunan, Jiangxi, Guangdong, Chongqing |

Note: see Appendix D for the geographic locations.

Even though it still remained unresolved to quantify the influences that the agricultural reforms and fiscal stimulus put on the growth path of agricultural labor productivity in China from our current results, further studies with latent growth curve modeling are possible to estimate the extent of the effects of the exogenous factors, such as reforms and macro policies, if the prefectural-level data on relative variables could be collected, which could provide more specific suggestions for the further improvements of the agricultural reforms, migration policies and government fiscal policies.

## 5. Conclusions

Our study firstly demonstrated that the Chinese agricultural labor productivity developed certainly as an increasing but non-linear trajectory. The quadratic and cubic curves might better describe the growing trend than the linear one, but still are not precise enough to fit the exact developing path of agricultural labor productivity in China. The piecewise model provided the best fitted trajectory to depict the development of agricultural labor productivity in China among all of the tested models. It breaks the entire trend into three linear pieces with two time points of 2004 and 2008. One breaking point at the year 2004 is the time when the agricultural tax and subsidy reforms had been introduced in the whole country. The other breaking point at the year 2008 is the first year to launch the 4-trillion RMB stimulus package to deal with the global recession. Meanwhile, the labor migration has fluctuated heavily in both years. Even though in the current study, we cannot quantify the direct correlation between the exogenous factors and growth in agricultural labor productivity, it is still empirically reasonable to analyze the potential effects of the agricultural reforms and labor migration on the promotion of agricultural labor productivity. This study can be a side view to confirm that the macro policies play important roles in the promotion of productivity in China besides the development of science and technology.

According to the previous analysis on the individual differences of growth paths, we could conclude that the changing trajectories of the agricultural labor productivity of the prefectural cities in China were following different paths. They started at different initial levels of agricultural labor productivity and also grew at different rates over time. To elaborate more specifically, the prefectures with lower efficiency in agricultural labor productivity have been growing at a relatively lower rate, while the ones with higher productivity were growing faster. The applications of the unspecified model and the piecewise model both confirmed this conclusion. It seemed that the Chinese regions are forming into two clubs in terms of agricultural labor productivity. Each club grows at its own pace and gets more disparate from the other, whereas there might be a convergence trend inside each group. The further analysis of the convergence estimation echoes the conclusions. The absence of σ-convergence indicates the enlarging gap of agricultural labor productivity across Chinese prefectures. The weak β-convergence implies that there might be some observations converging to the higher level; however, they cannot reverse the whole dispersion of agricultural labor productivity across different regions.

Moreover, the piecewise model shows that the disparity of agricultural labor productivity has been further enlarged after 2004 and 2008. In the first stage from 2000–2004, the growth trends of Chinese prefectures are relatively less dispersed even with different growth rates. In the second and third phases (2005–2008 and 2009–2013), there is no convergence in terms of agricultural labor productivity across Chinese regions, hence the distribution of regional agricultural labor productivity becomes more diverged compared to the previous stage. We can assume that the reform in 2004 has effectively promoted the growth of agricultural labor productivity, especially for the developed regions in the early stage, or in other words, the prefectures with better initial conditions of agricultural labor productivity obtained more benefits from the agricultural reforms. As far as we are concerned, the main reason might be that the developed regions normally possessed better circumstances and supporting facilities to implement the reforms more efficiently. Another possible reason might be the experimental reform before 2004 in some pilot regions with favorable agricultural conditions that provided advanced chances for them to make profits from the reforms. Meanwhile, it is also noteworthy to prevent aggressive government intervention in farming and to introduce the market mechanism into agricultural production. The 2008 stimulus package, however, has little effect on closing the regional gap in terms of agricultural productivity. Therefore, it is suggested to the local government and policy makers to attach more importance to the underdeveloped regions. However, blindly expanding the investment is not a wise choice to bridge the gap between the underdeveloped and developed regions. The key is to tailor different policies and investment bundles according to different local conditions and then to improve the macro circumstances for policy implementation. For instance, in the undeveloped regions with slow growth rates in agricultural labor productivity, the

critical problem is to enhance the rural infrastructure construction and agricultural technology at the same time. In the developed regions that with fast growth rates in agricultural labor productivity, the supplements for environmental protection and sustainable development in agricultural production shall be given more consideration; further related regulations shall be formulated to improve the existing agricultural subsidy system. In the main grain area, the EU's regime of green box agriculture to disintegrate subsidy from production can be taken as a reference to prevent production surplus.

**Acknowledgments:** The first author thanks the financial support received through the project Cambiamento istituzionale, crescita economica e sviluppo sociale funded by the autonomous Province of Trento. Authors also thank the great helps from Nica Claudia Calò, Barbara Barone, Roberto Fanfani, and Cristina Brasili.

**Author Contributions:** Peng Bin conceived and designed the experiments; Marco Vassallo performed the experiments and analyzed the data; both authors contributed analysis tools and wrote the paper.

**Conflicts of Interest:** The authors declare no conflict of interest.

## Appendix

### Appendix A. Changes of Agricultural Output and Inputs in China (1990–2013)

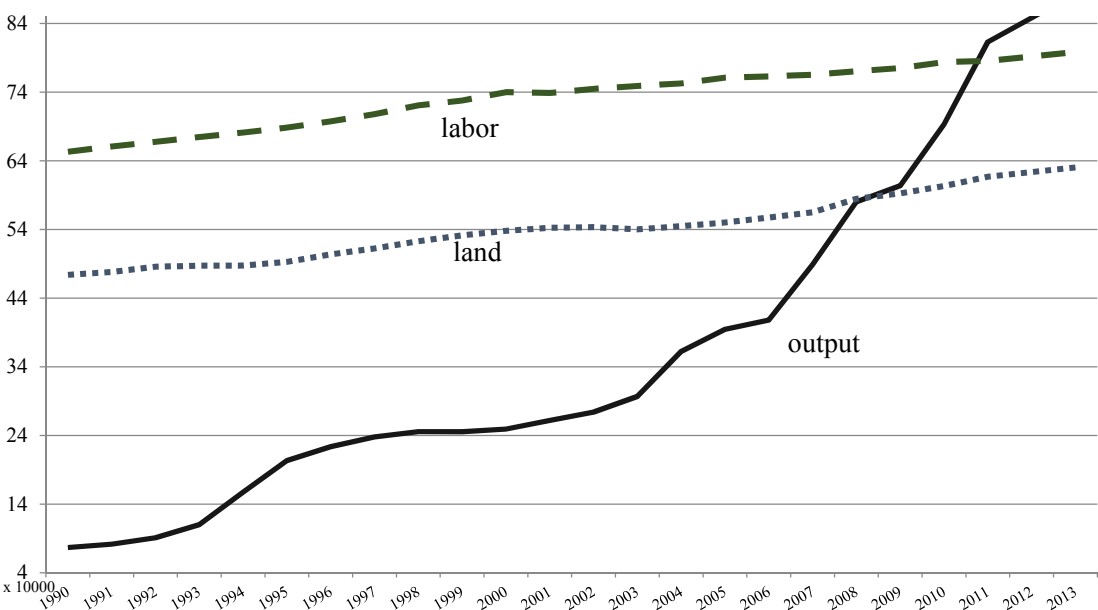

**Figure A1.** The absolute value of agricultural output and inputs. Source: data are collected from China's Statistical Yearbooks (1991–2014) [45]. The unit of agricultural output is 10,000,000 Yuan, of agricultural labor is 1000 persons and of land is 10 hectares. All of the values are real values with the base year of 1990.

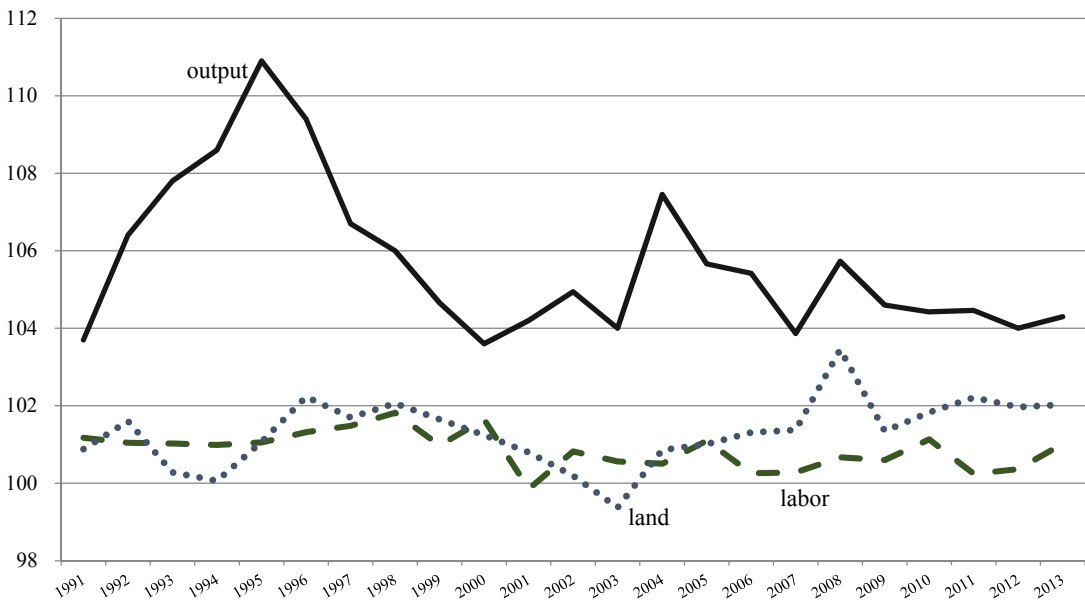

**Figure A2.** The index of agricultural output and inputs. Source: data are collected from China's Statistical Yearbooks (1991–2014) [45]. The *y*-axis is the index as the previous year = 100. All of the values are real values with the base year of 1990.

## Appendix B. The Modification Indices Suggest to Add an Error Covariance

**Table B1.** The Modification Indices Suggest to Add an Error Covariance.

| Year_1 | Year_2 | Decrease in Chi-square | New Estimate |
|--------|--------|------------------------|--------------|
| AP04 | AP03 | 281.1 | 87.97 |
| AP09 | AP06 | 105.3 | −119 |
| AP09 | AP08 | 120.8 | 231.96 |
| AP10 | AP08 | 123.2 | 259.73 |
| AP10 | AP09 | 216.6 | 672.01 |
| AP11 | AP09 | 163.9 | 679.26 |
| AP11 | AP10 | 157.3 | 715.82 |

## Appendix C. Convergence Estimation

This study conducts a simple measure of σ-convergence as employed in the study of Boyle and McCarthy (1999) [31]. The function is as follows:

$$\sigma = \frac{var\left(Y_{i,t}\right)/mean(Y_{i,t})}{var\left(Y_{i,0}\right)/mean(Y_{i,0})}$$

The methodology to test β-convergence was introduced originally by Baumol's (1986) [25] study of real convergence across economies. The function is as follows:

$$\frac{1}{T}\left[\ln\left(Y_{i,t}\right) - \ln\left(Y_{i,0}\right)\right] = \alpha + \beta\ln\left(Y_{i.0}\right) + \varepsilon_i$$

## Appendix D. Chinese Map of Administrative Districts

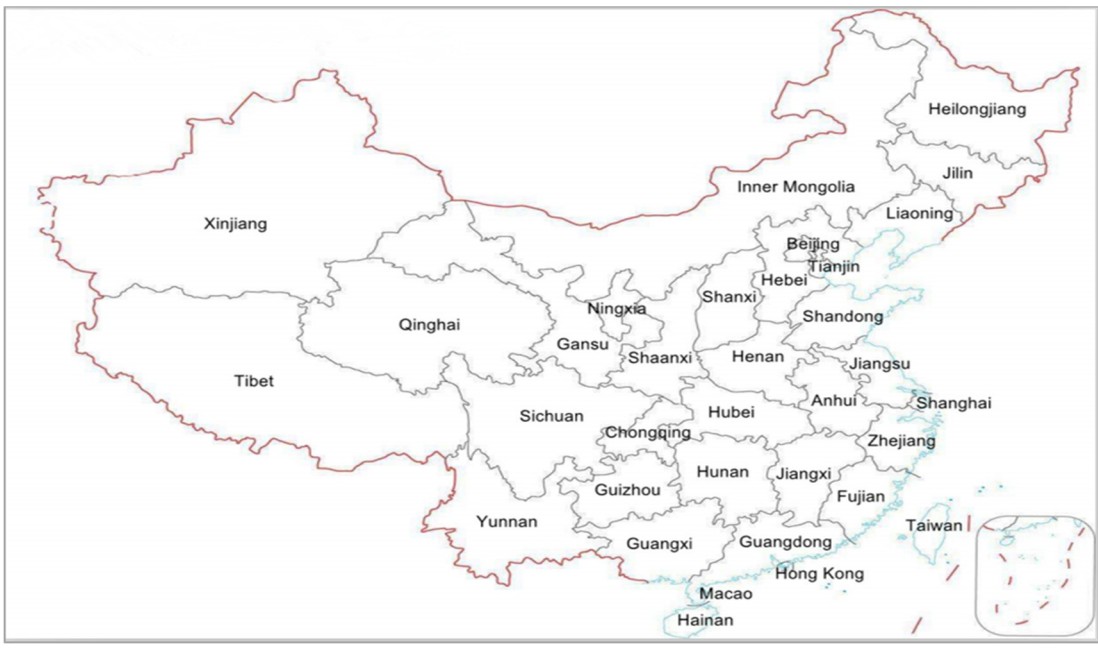

**Figure D1.** Chinese Map of Administrative Districts.

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
