# Peer review of "The Growth Path of Agricultural Labor Productivity in China: A Latent Growth Curve Model at the Prefectural Level"

_economies, doi:10.3390/economies4030013_

Round 1

Reviewer 1 Report

General comments

The authors apply Latent Growth Curve Model (LGCM) approach to characterize the growth path of China’s agricultural labor productivity between 2000 and 2010 using a balanced panel data. While the topic is interesting there are some major issues that need to be properly addressed first.

Data issue: The authors indicate their data source as “…an index of panel data on agricultural output and employment in the first sector of 287 prefectures from 2000 to 2010 is collected from the China city Statistical Yearbook (2001-2011)...” It is not clear what kind of index the authors are referring to. Are the annual agricultural output data are measured in their nominal value (at current price) or in real value (at constant price)? If it’s based on the real value then what kind of deflators are used in the estimation, national level or regional level since agricultural prices may vary across regions? Labor productivity needs to be measured using real value of agricultural output. Otherwise, it may reflect the changes in prices instead of productivity. This issue needs to be properly addressed.

Literature discussion: Most of the literature cited in the Introduction section and Discussion section are published five years ago. The authors may want to have some brief discussion on more recent literature regarding China’s productivity estimates and policy related issues.

Measurement issue: The authors rely on the Latent Growth Curve Modeling technique to study the trend growth of China’s agricultural labor productivity. It is not clear, however, why the authors rely on a dataset with only 10-year period to discuss labor productivity trend growth issue instead of using a dataset with longer time period if data is available. In addition, the authors may want to justify their approach by briefly discussing the advantage or shortcomings of their method comparing to other approaches in identifying the growth trend of China’s agricultural labor productivity.

Specific Comments

Please double check the citations and the references list as some citations are not included in the references.

Table 1: The title of column 2 is not clear. The numbers in that column seem to be cumulative proportions of total changes over time. Please be more specific.

Table 3: Each column title needs to be more specific.

Figure 3: It is not clear what the axis labels represent in figure 3. It needs to be noted.

Figure 4: Please be more specific regarding the range of slow, middle, and fast.

Appendix Figure A1: What is the definition of “absolute value” in this figure?

Appendix Figure A2: The figure presents the chain-linked index of output and input. It is not clear whether the output is measured in real value (at constant price) or nominal value (at current price). Please be more specific.

Author Response

We would like to thank the Reviewer for a very careful reading of the paper and a number of interesting, though challenging, suggestions and comments. Thanks to his/her remarks, the paper has improved under many important aspects.

We summarize below our reactions to the points raised by the Reviewers following the same order in the lists of comments. In general, we amended the paper in accordance with the replies below.

Data issue: The authors indicate their data source as “…an index of panel data on agricultural output and employment in the first sector of 287 prefectures from 2000 to 2010 is collected from the China city Statistical Yearbook (2001-2011)...” It is not clear what kind of index the authors are referring to. Are the annual agricultural output data are measured in their nominal value (at current price) or in real value (at constant price)? If it’s based on the real value then what kind of deflators are used in the estimation, national level or regional level since agricultural prices may vary across regions? Labor productivity needs to be measured using real value of agricultural output. Otherwise, it may reflect the changes in prices instead of productivity. This issue needs to be properly addressed.

-          We better specify the required items on agricultural labor productivity in the text. Sorry for the unclearness. The index is actually the measure of agricultural labor productivity, which is calculated by the agricultural output and labor. We now clarify this part with another way of expression. The agricultural output is deflated with the base year 2000. There is no price index at the prefectural level. For this reason we use the provincial price indices to deflate the nominal series. Hence, all the prefectures within each province are deflated by the same index. Although suboptimal, this helps to preserve the comparability in the two estimations.

Literature discussion: Most of the literature cited in the Introduction section and Discussion section are published five years ago. The authors may want to have some brief discussion on more recent literature regarding China’s productivity estimates and policy related issues.

-          The Reviewer raises a very valuable point. We would like to thank the Reviewer to recommend us to develop this part. We now include more recent literature in the introduction, as well as the government movements of the relevant policies in recent years.

Measurement issue: The authors rely on the Latent Growth Curve Modeling technique to study the trend growth of China’s agricultural labor productivity. It is not clear, however, why the authors rely on a dataset with only 10-year period to discuss labor productivity trend growth issue instead of using a dataset with longer time period if data is available. In addition, the authors may want to justify their approach by briefly discussing the advantage or shortcomings of their method comparing to other approaches in identifying the growth trend of China’s agricultural labor productivity.

-          The Reviewer raises an interesting point that we overlooked previously. We take the starting year as 2000 is mainly because of the entry of WTO, which influences the pattern of agricultural production in China. We take the reviewer’s advice, to extend the time period until 2013, which is the latest available data on prefectural level.

Specific Comments

Please double check the citations and the references list as some citations are not included in the references.  

Table 1: The title of column 2 is not clear. The numbers in that column seem to be cumulative proportions of total changes over time. Please be more specific.

Table 3: Each column title needs to be more specific.

Figure 3: It is not clear what the axis labels represent in figure 3. It needs to be noted.

Figure 4: Please be more specific regarding the range of slow, middle, and fast.

Appendix Figure A1: What is the definition of “absolute value” in this figure?

Appendix Figure A2: The figure presents the chain-linked index of output and input. It is not clear whether the output is measured in real value (at constant price) or nominal value (at current price). Please be more specific.

-          We would like to thank the reviewer’s scrutiny. We now correct all the issues in citations and tables and figures in the text. Figure 3 (now is Figure 4), the vertical axis indicates the growth rate of agricultural labor productivity; we now put it in the note. In Appendix A1, the “absolute value” means real value; we now change it to the better expression. In Appendix A2, they are measured in real value at constant price; we now better specify it in the text.

Reviewer 2 Report

First of all, the paper is very interesting. Despite that, the authors must improve the paper in order to get a better piece of work.

The most important issues to correct urgently:

One question that it is necessary to correct is the number of figures in the text.

Besides, the authors must define better the data how the agricultural labor productivity was calculated. The agricultural output is in current or constant prices? This could have strong implications in one case or another, especially in the first case. The context of this paper consists in growing agricultural prices, especially in the second half of this decade in the world panorama. So, if the agricultural output is in current prices, the growing prices will affect deeply the results. It must be clearer.

On the other hand, in the same line of text (pg. 4 line 3), the authors define the denominator of the agricultural labor productivity as the employed labors in the first sector. What does this first sector mean? Is it a synonymous of agricultural sector? If this “first sector” includes the fish workers, the agricultural labor productivity measures are not correct in the regions with sea access. In this case, it would be the primary sector labor productivity. The authors should also clarify if this sector included the forest products and labor.

It is necessary to concrete more the definition of the data, output and labor, that the authors used in this measurement, because this has strong implications.

In the map of agricultural labor growth in Chinese region, the legend must include which is considered a high growth, medium and low growth of this variable.

Issues to improve the paper:

The introduction is welldone, but it could be improved in some points. I have missed a review of the literature. The measurement of the agricultural labor productivity is an important theme in the literature, not only in the agricultural economics, but also in the economics literature. For example, there are two paper published in two important journals in economics recently that the authors omitted. These papers could be reinforcement about the relevance of this measurement of agricultural productivity: Gollin D, Lagakos D, Waugh ME (2014) Agricultural productivity differences across countries. American Economic Review 104(5):165–170; Gollin D, Lagakos D, Waugh ME (2014) The agricultural productivity gap. Quarterly Journal Economics 129(2): 939–993.

Other thing to improve the introduction would be writing the main objective of the paper. I think, this clear objective could improve the introduction about the target that the authors follow in this paper. In this version, this is relatively diffuse.

The authors could enrich the results’ part with reference to the concepts of convergence of Barro and Sala-i-Martin, beta and sigma convergence. The authors wrote about these concepts, for example pg. 6 lines 22-23 and pg. 9 lines 5-6, but they can point out the non existence in the case of Chinese prefectural level.

Maybe, the footnote 5 could go to the table 2’s note.

One of the most interesting issues of the paper is the political change in 2004. Perhaps, more detailed reasons of the reform would enrich the causes to promote it. Besides, the implications of this political reform are explained in the text, but more detail in these explanations would permit to understand better these changes.

As a suggestion for this paper or another, having the data of output, labor and land, it is also interesting to analyze the agricultural labor productivity explained by the agricultural land productivity and land-labor ratios. Maybe, some simple correlations between the growths of agricultural labor productivity vs land productivity and land-labor ratio could enrich this or future analysis.

Author Response

We would like to thank the Reviewers for a very careful reading of the paper and a number of interesting, though challenging, suggestions and comments. Thanks to their remarks, the paper has improved under many important aspects.

We summarize below our reactions to the points raised by the Reviewers following the same order in their lists of comments. In general, we amended the paper in accordance with the replies below.

The most important issues to correct urgently:

One question that it is necessary to correct is the number of figures in the text.

-          We would like to thank the reviewer’s scrutiny. We now specify this better in the text

Besides, the authors must define better the data how the agricultural labor productivity was calculated. The agricultural output is in current or constant prices? This could have strong implications in one case or another, especially in the first case. The context of this paper consists in growing agricultural prices, especially in the second half of this decade in the world panorama. So, if the agricultural output is in current prices, the growing prices will affect deeply the results. It must be clearer.

-          The reviewer rises a very useful point that we initially ignored. We now better specify the required items on agricultural labor productivity in the text. The agricultural output is deflated with the base year 2000. There is no price index at the prefectural level. For this reason we use the provincial price indices to deflate the nominal series. Hence, all the prefectures within each province are deflated by the same index. Although suboptimal, this helps to preserve the comparability in the two estimations.

On the other hand, in the same line of text (pg. 4 line 3), the authors define the denominator of the agricultural labor productivity as the employed labors in the first sector. What does this first sector mean? Is it a synonymous of agricultural sector? If this “first sector” includes the fish workers, the agricultural labor productivity measures are not correct in the regions with sea access. In this case, it would be the primary sector labor productivity. The authors should also clarify if this sector included the forest products and labor.

-          The reviewer is right. The first sector is a macro conception of agriculture. It indeed includes fishery. However, the data of output and employment are both collected on this level - the first sector. Hence, we assume the quotient of the two variables is a proxy to agriculture labor productivity. Meanwhile, they are the only available data on the prefectural level to calculate agricultural labor productivity. Therefore, we think this calculation is still helpful.

It is necessary to concrete more the definition of the data, output and labor, that the authors used in this measurement, because this has strong implications.

-          Yes, the Reviewer is correct. We now specify this better in the text.

In the map of agricultural labor growth in Chinese region, the legend must include which is considered a high growth, medium and low growth of this variable.

-          Yes, the Reviewer is correct. We now specify this better in the text.

Issues to improve the paper:

The introduction is welldone, but it could be improved in some points. I have missed a review of the literature. The measurement of the agricultural labor productivity is an important theme in the literature, not only in the agricultural economics, but also in the economics literature. For example, there are two paper published in two important journals in economics recently that the authors omitted. These papers could be reinforcement about the relevance of this measurement of agricultural productivity: Gollin D, Lagakos D, Waugh ME (2014) Agricultural productivity differences across countries. American Economic Review 104(5):165–170; Gollin D, Lagakos D, Waugh ME (2014) The agricultural productivity gap. Quarterly Journal Economics 129(2): 939–993.

-          The Reviewer raises a very valuable point. We would like to thank the Reviewer to recommend us to develop this part. We now include more recent literature in the introduction, as well as the government movements of the relevant policies in recent years.

Other thing to improve the introduction would be writing the main objective of the paper. I think, this clear objective could improve the introduction about the target that the authors follow in this paper. In this version, this is relatively diffuse.

-          Yes, the Reviewer is correct. We now specify this better in the text.

The authors could enrich the results’ part with reference to the concepts of convergence of Barro and Sala-i-Martin, beta and sigma convergence. The authors wrote about these concepts, for example pg. 6 lines 22-23 and pg. 9 lines 5-6, but they can point out the non existence in the case of Chinese prefectural level.

-          The Reviewer raises a valuable point which we overlooked previously. In the section 3.3, we improve the understanding of the growth pattern of Chinese agricultural labor productivity by adding the estimations of beta and sigma convergence. The results of convergence estimation also echo our analysis of LGCM.

Maybe, the footnote 5 could go to the table 2’s note.

-          Yes, the Reviewer is correct. Since we add three years into the dataset (following the suggestion from the other reviewer), the table context changes. We now improve this issue in the text.

One of the most interesting issues of the paper is the political change in 2004. Perhaps, more detailed reasons of the reform would enrich the causes to promote it. Besides, the implications of this political reform are explained in the text, but more detail in these explanations would permit to understand better these changes.

-          We improve this part in the text. The recent movements of the government in this agricultural reform are added. Since we add the data of recent three years, another break is observed in the curve. So we also introduce the macro policy of stimulus package.

As a suggestion for this paper or another, having the data of output, labor and land, it is also interesting to analyze the agricultural labor productivity explained by the agricultural land productivity and land-labor ratios. Maybe, some simple correlations between the growths of agricultural labor productivity vs land productivity and land-labor ratio could enrich this or future analysis.

-          The Reviewer raises a very interesting point. We would like to thank the Reviewer to recommend us to develop this part. However, the prefectural-level data in terms of farmland are only reported until 2007 in the China City Statistical Yearbook.

Round 2

Reviewer 1 Report

One specific comment:

Please double check the references list as some citations are not included in the references.  

Reviewer 2 Report

It is correct to be published and the authors have made a strong effort to consider my comments and suggestions.